# Associated Factors of *Mycobacterium Leprae* Infection among People with Leprosy in Kwale County

Vallerian Karani[1]*, Lameck Dierro[2], Aiban Rono[3], Martin Githiomi[3],
Wendy Rose Nkirote[3], Rhoda Pola[3], Stephen Joram Olubulyera[1],
James Marcomic Maragia[1], Dominic Ongaki[1], Fredrick Odhiambo[1], Maurice Owiny[1],
Dickens Onyango[4], Abade Ahmed[1]

**1** Kenya Field Epidemiology and Laboratory Training Program, Nairobi, Kenya, **2** Moi University, Eldoret, Kenya, **3** Division of National Tuberculosis, Leprosy and Lung Health Programme, Ministry of Health, Nairobi, Kenya, **4** Amref International University, Nairobi, Kenya

* vallerian08@gmail.com

## Abstract

### Background

Leprosy, a chronic bacterial disease caused by *Mycobacterium leprae*, is curable yet neglected. Approximately 200,000 new cases are reported globally each year, with India contributing 60%. In 2020, the African WHO regions had a leprosy burden of 14.9 per 1,000,000 population. Despite maintaining the global elimination target of <1/10000 population, Kenya reported a six-fold increase in cases from 2011 to 2021, with Kwale County contributed 24.3%. This study aimed to determine the factors associated with *m. leprae* infection.

### Methods/Findings

We conducted a 1:3 case-control study in the Kwale County from June-September, 2023. Case was any person who was diagnosed from any health facility and documented in the Leprosy register, including patients who recently completed treatment. Controls were defined as any person with no clinical signs and were a neighbor to a case, matched by residence, age group, and sex. Questionnaires were administered to both groups. Descriptive analysis was performed for continuous and categorical variables. Factors associated with Leprosy were evaluated using multivariable binary logistic regression. Stepwise backward elimination was used to build a final model; p-values of ≤0.05 were considered statistically significant.

A total of 65 cases and 195 controls were enrolled. The mean age was 55 years (SD ± 16) for the cases and 54 years (SD ± 15) for the controls (range:10–83 years). Among cases,56.6% (n = 37) were married, compared to 71.1% (n = 139) of controls. 55% (n = 36) of the cases and 41% (n = 81) of the controls were illiterate. Independent

**Data availability statement:** The authors declare that all data and related metadata underlying this study's findings are available at Dryad:https://datadryad.org.A unique, permanent digital object identifier: https://doi.org/10.5061/dryad.pg4f4qs14.

**Funding:** The author(s) received no specific funding for this work.

**Competing interests:** The authors have declared that no competing interests exist.

risk factors:family size ≥5 members (aOR=6.99, 95% CI: 2.71–18.06), family contact (aOR=4.33, 95% CI: 2.18–8.58), social contact (aOR=2.24,95% CI 1.16–4.32), and missing of a BCG vaccination mark (aOR=2.24,95% CI: 1.11–4.53).

## Conclusion/Significance

Large family size, family and social contacts, and missing BCG vaccination mark were associated with Leprosy.The Ministry of Health should sustain and expand BCG vaccination coverage among all eligible populations.

### Authors summary

Leprosy, though curable, continues to be a serious global concern, especially in regions with limited resources like sub-Saharan Africa. Our study in Kenya aimed to identify associated drivers that influence the spread of leprosy. We discovered that close contact within families and social groups, like distant relatives, coworkers, and friends, increased the probability of getting the disease. Lack of BCG vaccination, as shown by missing mark on the left forearm, may further facilitate the vulnerability. Overcrowded living conditions, especially in rural areas, was also associated with a higher proportion of leprosy.

Our findings suggest that countries should sustain and expand BCG vaccination programs to effectively eliminate leprosy, even though its effectiveness against leprosy is still debated. Improved surveillance efforts, such as contact tracing and door-to-door screenings, are extremely important in the fight against leprosy. Addressing overcrowding and food insecurity, which worsen the spread of the disease, is crucial as well. These strategies are vital in reducing transmission, enhancing community monitoring, and moving closer to the global goal of eliminating the disease. Our study highlights the need for targeted interventions in communities where leprosy is still persistent, ensuring that progress in fighting this neglected disease continues and grows.

### Introduction

Leprosy, is a neglected chronic bacterial infectious disease caused by *Mycobacterium Leprea,* though it is curable [1]. The disease is considered a major public health concern [2]. It manifests through anesthetic skin lesions, peripheral nerve enlargement, and the presence of acid-fast bacilli in skin smears [3]. Despite the lack of an effective vaccine, only 15% of the global population may contract the disease upon exposure [4,5]. Globally, more than 200,000 new leprosy cases are reported annually to the WHO, with India contributing 60% of these cases [6–8]. It is estimated that 3–4 million people suffer from disabilities resulting from leprosy [3,6]. In Africa, the continent had the second -highest proportion of new leprosy cases among WHO regions, with a leprosy burden of 14.9 per 1 million persons; Nigeria accounted for 40%, while Mozambique notified 2,065 new leprosy cases [7,8].

Since 1989, Kenya has maintained a global leprosy eradication goal at the national level, adhering to the WHO reduction target of less than one case per 10,000 people [1]. However, an unexplained regional surge of leprosy cases raises a significant concern. Over 80% of Kenyan counties have reported at least one or more active leprosy cases from 2011-2021 [9]. Many cases go unnoticed and undocumented [10,11]. In 2021, 16.2% of newly diagnosed patients had disability grade II against the global average of 6.7% [7,9]. This might be primarily attributed to either individual or healthcare system delays [12,13]. In 2019, approximately 7.5% of patients diagnosed in the country were children under 15 years, against the global target of less than 3% [7,9], an indication of active community transmission and fragile surveillance systems [11]. In Kwale County, a region predominantly inhabited by fishing communities, the situation is particularly dire, with half of the residents living below the poverty line. The county has consistently reported leprosy cases for the past decade, accounting for 24.3% of the national cases—significantly above the national average [11]. The high burden of leprosy in Kwale County suggests active transmission within the community, likely driven by the unique lifestyle and socio-economic challenges of the region. The coastal communities are characterized by fishing, communal living, and limited access to healthcare and education, creating an environment that facilitates leprosy transmission. Previous studies have identified various risk factors for leprosy, such as age, malnutrition, close contact with leprosy patients, and socio-economic conditions. However, these factors may manifest differently in this setting due to the distinct lifestyle and healthcare barriers present [7,14,15].

Leprosy remains a complex and poorly understood disease, likely due to its long incubation period [16], approximately three to seven years for Paucibacillary and up to seventeen years or more for Multibacillary leprosy [7,17]. People with the multibacillary form of leprosy are considered the main source of *m. leprae* transmission, primarily through droplets from coughing and sneezing [6,18].In addition, persons living in close contact with untreated multi-bacillary patients are at the highest chance of contracting the disease [19]. Some scholars argue that leprosy can also be spread through skin-to-skin contact, and interactions between humans and animals, although inhalation of fine bacteria droplets remains the largely suggested transmission pathway [20,21]. Most infected persons are asymptomatic [22,23], but some tend to develop visible signs like severe disability [23].

Leprosy disease predominantly affects the skin and peripheral nerves, but the more advanced multibacillary type of leprosy mainly affects the eyes, often due to delayed reporting and diagnosis [5,12,24–27]. The WHO emphasizes that grade II disability, which reflects more advanced impairments, is a more reliable indicator of the disease burden compared to prevalence rates [28,29]. This highlights the importance of closely monitoring the proportion of patients diagnosed with varying degrees of disability in leprosy-endemic populations, to better assess and address the community disease impact [3,24,25]. Leprosy, is understood to be a disease of poverty [16,30–32],but the specific socioeconomic attributes linked to *m.Leprae infection* are not well documented [15,19]. An elaborate understanding of the specific socioeconomic factors like low education attainment and low income for leprosy transmission is critical [8,33].

This study aims to identify the associated factors specific to Kwale County, focusing on the unique socio-economic, clinical, cultural, and environmental influences on leprosy transmission in this community. Understanding these factors is crucial for developing targeted interventions and improving leprosy control strategies in similar settings. In this regard, the study aims to determine the predictors associated with *m. Leprea* infection in the study setting.

## Materials and methods

### Ethics statement

We obtained ethical approval from Moi University Institutional Review and Ethics Committee (Reference number: IREC/405/2023) and the permit to conduct the study from the National Commission for Science, Technology and Innovation (Reference number: 896014). Permison to conduct the study was provided by the Kwale County Health Management Team, approval (Reference number:/CG/KWL/6/5/CDH/39/VOL..1/ (94). Both written and verbal consent was sought from

the study participants. The data confidentiality was maintained by replacing personal details with unique numbers so that it would be difficult to re-establish a link between the individual and his or her data. The investigators, the research assistants and Community Health Promoters signed the data confidentiality declaration form (S1 Text).

## Study design, site and populations

A 1:3 case-control study was conducted in rural Kwale County. Cases were identified from treatment registers and comprised leprosy patients who recently completed treatment. Controls were people with no signs and symptoms and were a neighbor to a case, matched to cases by residence, age group, and sex.

Kwale County is one of Kenya's coastal regions, bordering Taita Taveta and Kilifi counties, and Tanzania. The county has a population of approximately 820,199, and 51% being female with nearly half living in poverty, 47.4% compared to the national rate of 36.1% [34,35]. Kwale is divided into urban and rural areas, with Ukunda as the central town, featuring a mix of formal housing, slums, and makeshift structures lacking piped water or sanitation [35]. The county also faces health challenges, with an estimated 16,692 people living with HIV (2.9% prevalence) and 992 tuberculosis cases diagnosed in 2019 [34,35]. Kwale has five level 4 hospitals, all providing Leprosy treatment services [34].

The study was conducted between June and September, 2023, and included Kwale County residents who are leprosy patients, and those who recently completed treatment. Controls were people with no signs or symptoms of leprosy and matched by the same villages, age group (±10) and sex.

## Operational definitions

**Case**: Any person who presented with a hypopigmented skin patch lesions with loss of sensation, one or more enlarged nerves, and the presence of leprosy bacilli with or without positive skin smear for *Mycobacterium Leprea*, including patients recently discharged from treatment.**Control:** Residents from the same village as the case and were in the same age group (±10) and sex, with no signs or symptoms of disease. **Family contact:** Contact with a person suffering from leprosy in the same house and use of the same kitchen (nuclear family members living in the same house). **Social contact:** Contact with someone with leprosy but not from the same house and sharing a kitchen (extend family members, relatives including workplace, school, friends, and distant relative.

## Sample size, assumptions and sampling procedure

The Kelsey formula was used to compute the sample size [36]. A total of 65 cases and 195 controls were included in the study. Using close contact with known leprosy cases as the exposure of concern, approximately 60% of controls were considered exposed to *Mycobacterium Leprea* Infection. The least extreme odds ratio (OR) to be detected was 2.6 at a significance level of 95%.It was possible to detect variations in exposure to leprosy disease between cases and controls with a 95% confidence interval and a power of 80%.

**Selection of the cases.** The County Tuberculosis and Leprosy Coordinator provided the list of all the leprosy treatment facilities. A list of all leprosy patients including those who recently completed treatment were abstracted from the leprosy treatment registers (S1 Text). The lead investigators signed the data confidentiality declaration form (S2 Text) with all the treatment facilities to ensure patient's records were kept confidential and private. The data protection method of de-identification was applied to replace personal characteristics with unique numbers, making it impossible to re-establish a link between and his or her data. The Controls were screened using a checklist (S1 Table) to ensure that they had no visible signs or symptoms. Among the cases and controls, the mark on the left forearm was utilized to identify individuals who had received the Bacillus Calmette Guerin (BCG) vaccination. Consent was administered to adults above 18 years and assent was administered to children by seeking permission from the guardians at household level. (S3 Text). Only individuals who gave consent/assent were enrolled in the study where both written and verbal consent was sought. Patients who had completed the treatment course and were discharged from treatment registers were enrolled

sequentially until the required sample size was reached. The identified cases were traced to their respective villages and places of residence. The questionnaire (S4 Text) was used for interviews at the household level.

**Selection of the controls.** Controls were identified and followed from the exact village as cases. The village elder generated a list of all the households. A pen was spun to identify the first household from the direction where the tip of the pen points. The investigation team selected the first household following the direction of the pen's tip. In this regard, the first household was selected systematically; If there is no person on the homestead, the household was dropped(skipped), and the next household was selected, the pen was spun again and again as they moved to the next home from the direction of the pen's tip until all the households were covered. At home, permission was sought to conduct interviews in the household, irrespective of age. If there was no person on the homestead, the household was dropped, and the next household was selected. A list of people staying in that village was developed, the individuals in the age category of the case were identified, and if they were more than one, one was randomly selected through secret balloting by the use of papers marked yes or no. In the event of a refusal, the next home was chosen. This was repeated until each village had the required target of controls.

### Data collection, tools and variables

A questionnaire was developed to collect data [37]. Research assistants were responsible for interviewing cases and controls. The questionnaire was structured to capture all the necessary vital information that incorporates all the variables for the study, including selection of controls at the village level. Among the associated factors that were assessed and evaluated included socio-demographic, clinical, behavioral and socio-cultural, household and environmental factors.

### Data management and statistical analysis

Data were analyzed using using MS Excel and STATA version 16.1. Descriptive analysis used means (standard deviations) for continuous variables and counts (proportions) for categorical. Factors Associated with Leprosy were evaluated using logistic regression. During bivariate analysis, stratification was done to assess for confounding and effect modification.; variables with a (p<0.2), were incorporated in multivariate analysis. A step-wise backward elimination was used to identify the best fit model. Factors with ≤5, were considered statistical statistically significant Adjusted odds ratios were presented.

## Results and discussions

### Socio-demographic and clinical characteristics

We enrolled 260 participants, of which 65 were cases and 195 were controls. The mean age was 55 years (SD±16) for the cases and 54 years (SD±15) for the controls. The range for the cases was 10–83 years. More than half 56.6% (n=37) of the cases and most of the controls 71.1% (n=139) were married. More than fifty 58.5% (n=38) of the cases had family contact, compared to 21.4% (n=44) of controls. More than half 53.3% (n=35) of the cases had social contact, compared with only 32.3% (n=63) of controls. Most 86.8%(n=33) of the cases had close family duration for over five years, and 82% (n=29) of the cases had social contact, respectively, compared to 86.9% (n=39) and 77.6% (n=76) of the controls. Over forty-three 43.3% (n=28) of cases lacked the BCG mark on the left forearm, compared to only 16.5% (n=32) of controls (Table 1).

### Behavioral and socio-cultural characteristics

The majority, 80%(n=52) of cases had frequently changed bed linen, compared to over sixty, 66.6%(n=130) of controls. Only 20%(n=13) of cases frequently changed bed linen weekly.

Over Sixty-three 41(63.1%) of cases and more than half, 52.4% (n=102) of controls reported sharing their bed linen. Only 36.9%(n=24) of cases and 96 (49.2%) of controls shared other bed linens. Fishing was practiced by only

**Table 1. Social-demographic and clinical characteristics of the study participants, Kenya.**

| Characteristics | Disease status | Disease outcome | |
|---|---|---|---|
| | Totals(n=260) % | Cases(n=65) % | Controls(n=195) % |
| **Sex** | | | |
| Male | 147(56.5) | 39(60.0) | 108(55.4) |
| Female | 113(43.5) | 26(40.0) | 87(44.6) |
| **Age** | | | |
| **Mean, Standard deviation** | **55(±16)** | **55(SD±16)** | **54(SD±15)** |
| **Minimum and maximum** | 10-83 | 12-81 | 10-83 |
| 0-14 | 3(1.5) | 1(1.0) | 2(1.1) |
| 15-29 | 23(9.2) | 5(8.7) | 17(8.9) |
| 30-49 | 57(21.5) | 14(22.1) | 43(21.9) |
| 50 and above | 177(67.8) | 44(68.2) | 133(68.1) |
| **Marital status** | | | |
| Married | 176(67.7) | 37(56.9) | 139(71.3) |
| Single | 28(10.8) | 8(12.3) | 20(10.3) |
| Widowed | 56(21.5) | 20(30.8) | 36(18.5) |
| **Education level attained** | | | |
| None | 117(45.0) | 36(55.4) | 81(41.5) |
| Primary | 102(39.2) | 23(35.4) | 79(40.5) |
| Secondary | 36(13.9) | 4(6.1) | 32(16.4) |
| Tertiary | 5(1.9) | 2(3.1) | 3(1.5) |
| **Family contact** | | | |
| Yes | 82 (31.5) | 38 (58.5) | 44 (22.6) |
| No | 178 (68.5) | 27 (41.5) | 151 (77.4) |
| **Social contact** | | | |
| Yes | 98 (37.7) | 35 (53.9) | 63 (32.3) |
| No | 162 (62.3) | 30 (46.1) | 132 (67.7) |
| **Duration of family contact** | | | |
| Less than 5 years, | 10(12.2) | 5(13.2) | 5(11.4) |
| More than 5 years | 72(87.8) | 33(86.8) | 39(88.6) |
| **Duration of social contact** | | | |
| Less than 5 years | 22(22.4) | 6(17.1) | 22(22.4) |
| More than 5 years | 76(77.6) | 29(82.9) | 76(77.6) |
| **BCG mark** | | | |
| Present(yes) | 200(76.9) | 37(56.9) | 163(83.5) |
| Absent (No) | 60(23.1) | 28(43.1) | 32(16.5) |

16.6%(n=11) of cases and 13.3% (n=26) of controls over 5 years. Hunting in the forest was practiced by only 9.2%(n=6) of cases, and over 10.0%(n=20) of controls. More than half of the cases 58.8% (n=38) and 49.2% (n=97) of controls had knowledge of leprosy and its symptoms.

Among the cases, more than sixty 63.1% (n=41) of the cases were unemployed compared to 44.3% (n=115) of controls. Informal employment was reported in 32.23%(n=21) of cases and 53.8%(n=138) of controls. Food shortages were experienced by 76% (n=50) of cases and 64.1%(n=125) of controls over the past 5 years. Only 23%(n=15) of cases had three meals a day previously, 44.6%(n=29) had one meal, and 32.1%(n=21) had two meals in a day previously, compared to the control group where only 27.2% (n=53) could afford one meal a day (Table 2).

## Household and environmental characteristics

Majority of the cases, 89.2%(n = 58) and controls 82.2%(n = 161) resided in houses with mud/sand floors, compared to only 12.3%(n = 8) of cases and 14.4%(n = 29) of controls lived in houses with well-maintained floors (permanent structures). The average household size for cases was 7.6 individuals (SD ± 2.8), ranging from 2 to 17 members. Compared to controls with the average household size ranged from 2 to 19 individuals and 5.7 individuals (SD ± 2.8).Notably, the majority 90.0%(n = 59) of cases had household densities of five or more members, while slightly less than sixty 59.0% (n = 115) of controls, was reported by 55.5%(n = 36) of cases, compared to 46 22.2% (n = 46) of controls. Regarding contact with domestic animals, slightly less than half 49.2% (n = 32) of cases reported contact, versus 44% (n = 87) controls (Table 3).

## Multivariable binary logistic regression model

The analysis revealed associated factors with leprosy disease occurrence; the odds of being diagnosed with leprosy was 7 times higher among individuals from households with ≥5 members (aOR = 6.99; 95% CI: 2.71–18.06), than those from households with <5 members". The odds among those with close family contact were four times higher (aOR = 4.33; 95% CI: 2.18–8.58), compared to those who did not. Persons with social contact (aOR = 2.24; 95% CI: 1.16–4.32) and Missing the BCG mark on the left forearm (aOR = 2.24; 95% CI: 1.11–4.53) were associated with a twice higher odds of leprosy than those without social contact and BCG mark (Table 4).

## Discussion

In this study, large family size, contact with leprosy in the family, social contact, and lacking a BCG scar were independently associated with leprosy. Our results underscore the public health importance of large family size of five members and above, family contact, social contact, and the missing BCG mark on the left forearm. Our current study observed a discrepancy between the cases family size and the controls where among the cases households that had five members and above had a 6.99-fold chance of sustained *m. Leprea* infection. This was consistent with previous research in Ethiopia that highlighted an epidemiological relationship between overcrowded families and the vulnerability to leprosy disease [24,33], suggesting an increased contact timeframe for family members and the patients [38]. However, the findings by Wagner et al, (2015) in India, showed no association of large family size members with support of leprosy development [30].

This study revealed that among the cases that had close family contact with cases had a 4 times higher odds of developing leprosy disease. Our findings were in agreement with previous findings in Ethiopia, Brazil and Ghana that showed that prolonged individual household contact among Multibacillary leprosy patient may facilitates leprosy development largely due to the bacillus's clinical epidemiology [4,15,24], possibly suggesting the highest likelihood of *m. Leprea* infection might have happened before [8]. The findings highlight a national programmatic fragile surveillance system and undocumented and missed cases of leprosy in rural communities that seem to worsen the burden of the disease.

The association of social contacts with leprosy diagnoses has not been consistent in the literature. Our study found that the odds of being diagnosed with leprosy was twice higher among individuals with a social contact with leprosy than those without a social contact.This is consistent with findings from a study done in Ethiopia by Bekala et al,2021, that tend to confirm that social contacts for leprosy are vulnerable to *m. Leprea* development [16,24,31]. In contrast to our current data, Ofusu et al, (2011) reported that there was no connection between social contact and the leprosy disease development [4],this might have been attributed to the selection design of the controls. Our findings, along with those from Ethiopia suggest that investigation of leprosy contacts could be broadened to include social contacts- distant and immediate relatives, social friends, and even workmates.

**Table 2. Behavioral and socio-cultural characteristics of the study Participant, Kwale county Kenya.**

| Variable Factor | Disease status | Disease outcome | |
|---|---|---|---|
| | Totals(n = 260) % | Cases(n = 65) % | Controls(n = 195) % |
| **Frequency of changing bed linen** | | | |
| Less than a Week | 78(30.0) | 13(20.0) | 65(33.1) |
| More than a Week | 182(70.0) | 52(80.0) | 130(66.9) |
| **Sharing own bed linen** | | | |
| Yes | 143(55.0) | 41(63.1) | 102(52.3) |
| No | 117(45.0) | 24(36.9) | 93(47.7) |
| **Duration of sharing own bed linen** | | | |
| More than 5 years | 122(85.3) | 37(90.2) | 85(83.3) |
| Less than 5 years | 21(14.7) | 4(9.8) | 17(16.7) |
| **Sharing other bed linen** | | | |
| Yes | 120(46.2) | 24(36.9) | 96(49.2) |
| No | 140(53.8) | 41(63.1) | 99(50.8) |
| **Duration of sharing other bed linen** | | | |
| More than 5 years | 96(80) | 19(79.2) | 77(80.0) |
| Less than 5 years | 24(20) | 5(20.8) | 19(20.0) |
| **Fishing, 5 years previously.** | | | |
| Yes | 37(14.6) | 11(16.9) | 26(13.8) |
| No | 222(85.4) | 54(83.1) | 168(86.2) |
| **Hunting, 5 years previously** | | | |
| Yes | 26(10) | 6(9.2) | 20(10.0) |
| No | 234(90) | 59(90.8) | 175(90.0) |
| **Regular bathing in the open** | | | |
| Open bathing place | 67(91.8) | 22(78.6) | 45(84.9) |
| River | 1(1.4) | 1(3.3) | 2(3.4) |
| Dam | 5(6.8) | 5(17.8) | 6(11.7) |
| **Ever experienced food shortage** | | | |
| Yes | 175(67.3) | 50(76.9) | 125(64.1) |
| No | 85(32.7) | 15(23.1) | 70(35.9) |
| **Employment** | | | |
| Informal Employment | 138(53.08) | 21(32.3) | 117(60.0) |
| Formal Employment | 7(2.7) | 3(4.6) | 4(2.7) |
| Unemployed | 115(44.23) | 41(63.08) | 74(37.9) |
| **Main meals** | | | |
| One meal | 82(31.5) | 29(44.6) | 53(27.2) |
| Two meals | 93(35.8) | 21(32.3) | 72(36.9) |
| Three Meals | 85(32.7) | 15(23.1) | 70(35.9) |
| **Knowledge of leprosy disease** | | | |
| Yes | 134(51.5) | 38(58.5) | 96(49.2) |
| No | 126(48.5) | 27(41.5) | 99(50.8) |

Another study in Ethiopia, by Urgesa et al, (2021) revealed that prolonged delay in leprosy case detection among families in endemic settings might sustain the leprosy transmission [39], especially in families with elderly contacts [8].

Findings from this study revealed that among the cases who did not have the Bacillus Calmette Guerin mark on the left forearm had twice odds of enhanced *m. Leprea* infection. Our results are in agreement with previous findings from Brazil

**Table 3. Household and environmental characteristics of study participants, Kwale county Kenya (n = 260).**

| Variable Factors | Disease status | Disease outcome | |
|---|---|---|---|
| | Totals(n = 260) % | Cases(n = 65) % | Controls(n = 195) % |
| **Type of floor** | | | |
| Mud floor | 219(82.4) | 58(89.2) | 161(82.5) |
| Cemented/tiled | 41(17.6) | 7(11.8) | 34(17.5) |
| **Type of housing** | | | |
| Permanent house | 37(14.2) | 8(12.3) | 29(14.1) |
| Grass thatched house | 223(85.8) | 57(87.7) | 166(85.9) |
| **Family size (household density)** | | | |
| <5 members | 86(33.1) | 6(9.2) | 80(41.0) |
| ≥5 members | 174(66.7) | 59(90.8) | 115(59.0) |
| **Animals in the compound,** | | | |
| Yes | 119(45.8) | 32(49.2) | 87(44.6) |
| No | 141(54.2) | 33(51.8) | 108(55.4) |

**Table 4. Multivariable binary logistic regression model for associated factors with *Mycobacterium leprae* infection Kwale county, Kenya.**

| Risk Factors | Disease status | | Disease outcome | | Disease outcome | |
|---|---|---|---|---|---|---|
| | Cases(n = 6 5) % | Controls(n = 195) % | cOR (95% CI) | P-value | aOR (95% CI) | P-value |
| **Family size** | | | | | | |
| ≥5 members | 174(66.7) | 59(90.8) | **6.84(2.82-16.61)** | **<0.001**\*\* | **6.99(2.71-18.06** | **<0.001**\*\* |
| <5 members | 86(33.1) | 6(9.2) | Reference. | | Reference. | |
| **Family contact** | | | | | | |
| Yes | 38(58.5) | 82(31.5) | **4.83(2.66-8.78)** | **<0.001**\*\* | **4.33(2.18-8.58)** | **<0.001**\*\* |
| No | 27(41.5) | 178(68.5) | Reference | | Reference | |
| **Social contact** | | | | | | |
| Yes | 35(53.9) | 98(37.7) | **2.44(1.38-4.33)** | **0.002**\*\* | **2.24(1.16-4.32)** | **0.016**\*\* |
| No | 30(46.1) | 162(62.3) | Reference | | Reference | |
| **BCG mark** | | | | | | |
| Absent (No) | 32(16.4) | 28(43.1) | **3.85(2.07-7.17)** | **<0.001**\*\* | **2.24(1.11-4.53)** | **0.024**\*\* |
| Present (Yes) | 163(83.6) | 37(56.9) | Reference | | Reference | |
| **Frequency of changing bed linen** | | | | | | |
| >Week | 52(80.0) | 182(70.0) | **2.69(1.02-3.94)** | **0.045**\*\* | NS | NS |
| ≤Week | 13(20.0) | 78(30.0) | Reference. | | | |
| **Main meals** | | | | | | |
| One meal | 29(44.6) | 53(27.2) | **2.55(1.25-5.24)** | **0.011**\*\* | NS | NS |
| Two meals | 21(32.3) | 72(36.9) | 1.36(0.65-2.85) | 0.414 | NS | NS |
| Three Meals | 15(23.1) | 70(35.9) | Reference | | | |

**\*P<0.2; \*\*P<0.05**; NS = Variable Not Significant, dropped from the final model as per the technique.

**Abbreviations**; cOR =Crude Odds Ratio, aOR=Adjusted Odds Ratio; CI = Confidence Interval 266

that showed that individuals who did not receive any dose of the Bacillus Calmette Guerin vaccine had a higher probability of leprosy disease when compared to those who did [5]. We discovered a discrepancy between cases and controls regarding the missing BCG mark on the left forearms; only 16.9% of the controls had no BCG marks, compared with the 43.1% of the cases. Although our data showed an association with the missing BCG mark on the left forearm, more

research is required to understand the possible association between BCG vaccination and leprosy disease. In addition, previous research documents that BCG vaccination partly offers protection against leprosy disease, with correspondence of approximately 50% effectiveness in protection [4,5].

We found out that persons who skipped meals to deal with hunger often ate one meal (46%) or even slept hungry, were two times more likely to develop leprosy disease. Comparable findings were reported by Oktaria et al, (2018) in Brazil, where the cases with low Body Mass Index had higher odds of contracting leprosy disease [16,31].This finding seems to attribute that persons with little income may have no money to spend on food making them vulnerable to *m. Leprea* infection. [24,39].However, Wagenaar et al, (2015) in India, argue that additional research is needed to determine the direct link between the host response to latent leprosy disease severity and hunger [30,31].Our current results reported a low frequency of changing bed linen had a 2.7-fold higher odds of *m. Leprea* infection. This is consistent with findings in Brazil and Ethiopia on social determinants of leprosy cases by Nery et al, (2019),which showed that inadequate access to basic clean water supply, poverty, and poor personal hygiene -such as irregular changing of bedlinens were associated with the development leprosy disease [15,18,40,41]. Previous results documented by Oktaria et al, (2018),hence increasing the family risk infection [16].These findings illustrate the important associated factors and facilitators for leprosy disease that further require extensive research in rural populations.

We did not show an association with gender, age, hunting, fishing, and domestic animal interaction as likely to influence the development of leprosy disease. However, other studies have positively linked leprosy to the elderly age group and male gender, particularly those living in resource-constrained settings [4,8,20,22,25,38,41]. Our findings also validate the existing information that *m. Leprea* primarily infects humans.

One of the limitations of this study was the possible recall bias among the study participants due to the study design. The study considered recall bias when creating the questionnaire and pre-testing. While a BCG mark on the left forearm was used in the study as a substitute to establish immunity against leprosy disease, it introduced bias since we were not able to distinguish those who got vaccinated but did not develop the scar from those who never got the vaccine. Despite these limitations, the study had several strengths; it employed a frequency-matched case-control study, a reliable study design for rare conditions. This study also adopted a 1:3 case-control design to increase the power of the study to 80%

## Conclusion and significance

Our results conclude that large family sizes of five members and above in rural populations was a significant associative factor for *m. Leprea* infection. It's associated with facilitating regular close contact, which increase the *m. Leprae* infection among rural populations. Close family and social contact highlights significant factors in the *m. leprae* infection. Social interactions within nuclear families, distant relatives, workmates and friends can sustain the transmission of *m. Leprea* infection, hence the need to implement tailored household interventions. The missing BCG mark on the left forearm suggests a lack of BCG vaccination, is highly associated with sustained *m. leprae* infection. The findings illustrate the significance of BCG vaccination in partly preventing *m. leprae* infection.

Ministry of Health in Kenya should enhance household contact surveillance for all leprosy cases besides household family contacts and involve social contacts like workmates, distant relatives and social friends as a strategy to eliminate leprosy transmission in the community. In addition, the ministry should also sustain and expand the uptake of BCG vaccination coverage among all eligible persons.

## Supporting information

**S1 Table. Screening checklist for controls.**
(DOCX)

**S1 Text. Data abstraction tool.**
(DOCX)

**S2 Text. Data confidentiality agreement.**
(DOCX)

**S3 Text. Consent and assent forms.**
(DOCX)

**S4 Text. Study questionnaire.**
(DOCX)

## Acknowledgments

We would like to acknowledge the study respondents for participation and gratitude in study. We are grateful to the research assistants for their contribution in data collection, and the County Government Kwale for the administrative support. We equally extend our gratitude to the Health Care workers, Community Health Promoters and the Village elders for their enormous support and contribution.

## Author contributions

**Conceptualization:** Vallerian Karani.

**Data curation:** Vallerian Karani, Aiban Rono, Fredrick Odhiambo.

**Formal analysis:** Lameck Dierro, Stephen Joram Olubulyera.

**Investigation:** Rhoda Pola.

**Methodology:** Vallerian Karani, Abade Ahmed.

**Project administration:** Maurice Owiny.

**Software:** Martin Githiomi.

**Validation:** James Marcomic Maragia.

**Visualization:** Dominic Ongaki.

**Writing – original draft:** Vallerian Karani.

**Writing – review & editing:** Wendy Rose Nkirote, Dickens Onyango.

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
