## [Decision Letter · Decision Letter 0]

15 May 2025

Associated Factors of Mycobacterium Leprae Infection among People with Leprosy in Kwale County, Kenya

Dear Dr. Karani,

Thank you for submitting your manuscript to PLOS Neglected Tropical Diseases. After careful consideration, we feel that it has merit but does not fully meet PLOS Neglected Tropical Diseases's publication criteria as it currently stands. Therefore, we invite you to submit a revised version of the manuscript that addresses the points raised during the review process.

Please submit your revised manuscript within 60 days Jul 14 2025 11:59PM. If you will need more time than this to complete your revisions, please reply to this message or contact the journal office at plosntds@plos.org. Please include the following items when submitting your revised manuscript:

We look forward to receiving your revised manuscript.

Kind regards,

Mathieu Picardeau

Section Editor

Shaden Kamhawi

co-Editor-in-Chief

Paul Brindley

co-Editor-in-Chief

**Additional Editor Comments :**

As suggested by the reviewers, it is necessary to provide a clearer description of the control group used in this study.

**Journal Requirements:**

At this stage, the following Authors/Authors require contributions: Vallerian Karani, and Fredrick Odhiambo. Please ensure that the full contributions of each author are acknowledged in the "Add/Edit/Remove Authors" section of our submission form.

3) Thank you for stating "Consent was administered to adults above 18 years and assent was administered to children by seeking permission from the guardians at household level." Please state whether the consent obtained from the guardians is verbal or written.

4) Thank you for stating "The authors declare that all data and related metadata underlying this study's findings are available and will be provided in a public repository or included within the submitted article, ensuring transparency and accessibility for further research and verification." We noted that the dataset submitted to the Dryad database is currently private. Please note that, though access restrictions are acceptable now, your entire minimal dataset will need to be made freely accessible if your manuscript is accepted for publication.

Should your submission be accepted, we will require the following information in your Data Availability Statement: 

1. The DOI provided by Dryad 

2. The citation for your data package in the reference section of your manuscript 

3. The citation for your data package in the methods section 

**Comments to the Authors:**

**Please note that two reviews are uploaded as attachments.**

**Reviewers' Comments:**

Reviewer's Responses to Questions

**Key Review Criteria Required for Acceptance?**

**Methods**

-Are the objectives of the study clearly articulated with a clear testable hypothesis stated?

-Is the study design appropriate to address the stated objectives?

-Is the population clearly described and appropriate for the hypothesis being tested?

-Is the sample size sufficient to ensure adequate power to address the hypothesis being tested?

-Were correct statistical analysis used to support conclusions?

-Are there concerns about ethical or regulatory requirements being met?

Reviewer #1: The objectives are clear;

Good methodology: The selection of controls was well done. It's a good idea to choose cases, patients who have just finished treatment or who are currently undergoing treatment

However, describe more who the control is; is it a member of the same household, a neighboring household in the same plot or another plot?

Reviewer #2: - The method described for selecting control households using a "pen spin" is arbitrary, non-standard, and introduces significant potential for selection bias. This undermines the internal validity of any results.

- Definition of cases is confusing. It is not clear what the diagnostic criteria is. Is it clinical or laboratory confirmed or both? What time frame did the authors use for “recently completed treatment”?

- The method for confirming controls had "no signs or symptoms" is not specified. Was this based on self-report or clinical examination?

- Key methodological details are missing or unclear, such as the specific age-matching criteria, validation/pre-testing of the questionnaire, and handling of refusals during recruitment.

- The reliance on a BCG scar as a proxy for vaccination without sufficient acknowledgment of its limitations in the methods section is problematic.

- The statistical analyses subsection is badly written. The rationale for using stepwise backward elimination for model building is not provided, and potential issues with this method are not acknowledged. Why was this method even considered if this was not a prediction study?

Reviewer #3: -Are the objectives of the study clearly articulated with a clear testable hypothesis stated?

An elaborate understanding of the specific socioeconomic factors like low education attainment and low income for leprosy transmission is critical. This seems interesting but will be a challenge to determine the predictors predictors associated with m. Leprea infection in the study setting as there so many determinant interconnected

-Is the population clearly described and appropriate for the hypothesis being tested?

• “ Cases were identified from treatment registers and comprised leprosy patients who recently completed treatment”. Need to define more clearly patients who recently completed treatment. What does it mean recently?

• Poverty line. Provide figures and sources for the country/location

• Please provide % of population living <5k; 5-10km; >10km from health facilities providing leprosy treatment of the study area

-Are there concerns about ethical or regulatory requirements being met?

I think yes

**Results**

-Does the analysis presented match the analysis plan?

-Are the results clearly and completely presented?

-Are the figures (Tables, Images) of sufficient quality for clarity?

Reviewer #1: In a case-control study like this one, we look for previous exposure factors that are associated with the disease. It will be necessary to explain that it is the fact of having previously been a family contact or a social contact that is significantly associated with leprosy. Similarly, it should be said that non-vaccination with BCG is associated with leprosy and not the absence of a BCG scar because the previous exposure was the vaccination which is evidenced by the scar

Reviewer #2: - There are numerous inconsistencies between the text descriptions and the data presented in the tables.

- Percentages in tables do not always appear logical or sum correctly. In Table 2, under the "Employment" variable, if you sum these percentages for the Controls: 60.0% + 2.7% + 37.9% = 100.6

Reviewer #3: -Does the analysis presented match the analysis plan?

Yes

-Are the results clearly and completely presented?

Yes, but I suggest if possible that factors associated to geographic and financial access to health care are taken into account.

-Are the figures (Tables, Images) of sufficient quality for clarity?

Yes

**Conclusions**

-Are the conclusions supported by the data presented?

-Are the limitations of analysis clearly described?

-Do the authors discuss how these data can be helpful to advance our understanding of the topic under study?

-Is public health relevance addressed?

Reviewer #1: The limitations of analysis are not clearly described

This is an interesting study in a particular context in Kenya which confirms that family and social contacts are more likely to develop leprosy in a context of poverty and promiscuity

Reviewer #2: There are flaws in the study design, methodology, data presentation, and interpretation. The conclusions drawn are not adequately supported by the methodology and data presented and the manuscript, in its current form, does not meet the scientific rigor expected for publication.

Reviewer #3: -Are the conclusions supported by the data presented?

Yes. But in summary, all the results were not presented. It focused only result associate to BCG. Please elaborate.

-Are the limitations of analysis clearly described?

Yes

-Do the authors discuss how these data can be helpful to advance our understanding of the topic under study?

Yes, but we need to add the determinants related to geographic and financial access to health services

-Is public health relevance addressed?

Yes, it contributes if the recommendation are more elaborated structuring by Individual, community, environment, health system responsibilities to addressing the transmission of m. leprae

**Editorial and Data Presentation Modifications?**

Reviewer #1: Minor revision

Reviewer #2: (No Response)

Reviewer #3: See in the attachment

**Summary and General Comments**

Reviewer #1: See file attached

Reviewer #2: The manuscript frequently uses "Mycobacterium Leprea" instead of the correct Mycobacterium leprae. This basic error raises concerns about overall scientific rigor.

Reviewer #3: (No Response)

PLOS authors have the option to publish the peer review history of their article (what does this mean? ). If published, this will include your full peer review and any attached files.

**Do you want your identity to be public for this peer review?** For information about this choice, including consent withdrawal, please see our Privacy Policy .

Reviewer #1: **Yes: ** Gilbert Batista

Reviewer #2: No

Reviewer #3: No

**Figure resubmission:**

**Reproducibility:**



---

## [Decision Letter · Decision Letter 1]

10 Sep 2025

Factors Associated with New Leprosy Diagnosis in Kwale County, Kenya

Dear Dr. Karani,

Thank you for submitting your manuscript to PLOS Neglected Tropical Diseases. After careful consideration, we feel that it has merit but does not fully meet PLOS Neglected Tropical Diseases's publication criteria as it currently stands. Therefore, we invite you to submit a revised version of the manuscript that addresses the points raised during the review process.

Please submit your revised manuscript within 60 days Oct 10 2025 11:59PM. If you will need more time than this to complete your revisions, please reply to this message or contact the journal office at plosntds@plos.org. Please include the following items when submitting your revised manuscript:

We look forward to receiving your revised manuscript.

Kind regards,

Georgios Pappas

Section Editor

Mathieu Picardeau

Section Editor

Shaden Kamhawi

co-Editor-in-Chief

Paul Brindley

co-Editor-in-Chief

**Journal Requirements:**

We ask that a manuscript source file is provided at Revision. Please upload your manuscript file as a .doc, .docx, .rtf or .tex. If you are providing a .tex file, please upload it under the item type u2018LaTeX Source Fileu2019 and leave your .pdf version as the item type u2018Manuscriptu2019.

**Reviewers' Comments:**

Reviewer's Responses to Questions

**Key Review Criteria Required for Acceptance?**

**Methods:**

-Are the objectives of the study clearly articulated with a clear testable hypothesis stated?

-Is the study design appropriate to address the stated objectives?

-Is the population clearly described and appropriate for the hypothesis being tested?

-Is the sample size sufficient to ensure adequate power to address the hypothesis being tested?

-Were correct statistical analysis used to support conclusions?

-Are there concerns about ethical or regulatory requirements being met?

Reviewer #1: - The objectives of the study are clearly stated

- The design of this case-control study is appropriate for the objectives set, as the prevalence is low and the incubation period of the disease is long

- The population is clearly described for both cases and controls

- The sample size is sufficient

- Correct statistical analyses were used to support the conclusions

Reviewer #2: (No Response)

Reviewer #3: No comment

Reviewer #4: Objectives are clear, but the hypothesis should have been articulated more explicitly. The design is appropriate but has potential biases that should be acknowledged more explicitly. The population is well described and mostly appropriate, though definitions of “cases” and reliance on proxy measures should be refined. Adequate for primary objectives but underpowered for less common exposures. Appropriate methods were applied, but reliance on stepwise selection weakens robustness. Ethical standards are met.

**Results:**

-Does the analysis presented match the analysis plan?

-Are the results clearly and completely presented?

-Are the figures (Tables, Images) of sufficient quality for clarity?

Reviewer #1: The results are clearly presented, However, there are some sentences that need to be reworded.

Reviewer #2: (No Response)

Reviewer #3: No comment

Reviewer #4: Mostly consistent, but minor discrepancies reduce clarity. Tables are comprehensive but crowded. Some results (e.g., recall bias effect) are mentioned but not quantified. Tables are informative, but:Some percentages do not sum exactly (rounding errors).Formatting could improve readability (e.g., separating crude and adjusted models).No graphical representation (forest plots, bar graphs), which would aid interpretation.

**Conclusions:**

-Are the conclusions supported by the data presented?

-Are the limitations of analysis clearly described?

-Do the authors discuss how these data can be helpful to advance our understanding of the topic under study?

-Is public health relevance addressed?

Reviewer #1: The data supported the conclusions but there are some sentences that need to be reworded.

Reviewer #2: (No Response)

Reviewer #3: No comment

Reviewer #4: Yes. The authors conclude that larger household size, family/social contact, and lack of BCG vaccination are major risk factors for new leprosy diagnoses . These are directly supported by the logistic regression findings.However:

The causal language (“predictors”) should be tempered, since this is an observational study.

Alternative explanations (e.g., genetic susceptibility, subclinical cases) are not addressed. Potential overmatching of controls is not acknowledged. Stepwise regression’s limitations are not adequately discussed.hey suggest that findings inform targeted interventions, such as household-level screening and vaccination campaigns . The study contributes to contextualizing global leprosy knowledge into a local Kenyan setting, advancing understanding of socio-economic risk markers. This provides actionable insights for the Kenyan national program and aligns with WHO leprosy strategies.

**Editorial and Data Presentation Modifications?**

Reviewer #1: I recommend "Accept" after minor corrections

Reviewer #2: (No Response)

Reviewer #3: No comment

Reviewer #4: Major revision required

**Summary and General Comments:**

Reviewer #1: This study identified factors associated with the diagnosis of new cases of leprosy.

The case-control study methodology is justified because the prevalence is very low, and the incubation period of the disease is long.

The study is interesting because the authors evaluated many socio-economic, environmental, and social factors and the study confirm the results already known, namely that family and social contacts of leprosy cases were at greater risk of developing the disease and that BCG vaccination was a protective factor.

Reviewer #2: Introduction

- Authors should familiarise with the standard binomial convention for naming micro-organisms. This was a comment in the previous round of review and should be taken seriously. The species epithet should be in the lower case.

- Authors should provide estimates of leprosy incidence/prevalence in Kenya in the introduction

-Line 78: Greater caution should be exercised in causally attributing occurrence of disability in leprosy to health system factors.

-Line 83: More clarity is needed in the phrase "consistently reported leprosy cases for the past decade". Does this mean leprosy cases were reported every year for the past decade?

-Line 85: National average should be reported for context.

-Line 86: More caution in causal attributions. Moreover, it is not clear how these ecological factors facilitate transmission. More clarity is needed.

Line 91: "However, these factors may manifest differently in this setting due to the distinct lifestyle and healthcare barriers" Not sure what this means, or what point is being made here.

Line 96: The genus name/abbreviation should be capitalised

Line 98: Would it be better to say "more susceptible" rather than "at the highest chance"?

Methods

- Line 135-136: Should be revised. Sounds confusing, as though leprosy is the treatment regimen.

- The text mentions cases were "treated with leprosy based on clinical/laboratory confirmation" but then states that Kenya "mainly diagnoses clinically and epidemiological criteria, due to limited slit-skin smear use." This contradiction needs clarification.

- I have read Clinical, laboratory, epidemiological criteria more then twice and still cannot decipher what this criteria is. The authors should be more direct and specific about what point they are trying to pass. If the cases were selected based on being on treatment and the authors have no access to the diagnostic criteria utilised, then the authors can succinctly mention how the cases were identified for their study.

- Uncertain why the case and control definitions are repeated in the study design section.

Operational definitions: This section is redundant and repeating what has already been said.

Sample Size, Assumptions and Sampling Procedure

- Line 178-181: This sentence is confusing "Using close contact with known leprosy cases as the exposure of concern, approximately 60% of controls were considered exposed to Mycobacterium Leprae Infection." How was exposure defined? Are the authors saying not all controls were considered exposed?

- The authors state that "approximately 60% of controls were considered exposed to Mycobacterium Leprae Infection". This is a very specific and high estimate for a general population sample. The manuscript provides no source or rationale for this figure.

- "Least extreme odds ratio" is confusing. Do you mean minimum detectable OR?

-Why choose OR of 2.6 as a clinically meaningful benchmark?

- The phrasing suggests this might be a post-hoc power calculation, which is methodologically questionable.

- Mentioning both "significance level of 95%" and "95% confidence interval" is repetitive and unnecessary.

- The case and control selection procedure should come much earlier.

- Why do the authors have study limitations in the methods?

Data management and statistical analysis

- Authors have not addressed statistical comments from the previous round of review.

- Maybe this was not clear in the previous round of review, stepwise methods are criticized because they use a purely algorithmic approach to determine model composition, which can lead to the exclusion of important confounders or the inclusion of statistically significant but clinically irrelevant variables.

- For an explanatory study like this, a more robust approach, such as purposeful selection based on prior epidemiological knowledge and assessing changes in estimates, would have been more appropriate.

- The authors state they frequency-matched controls to cases on sex, age-group (±10 years), and village. However, the logistic regression model used for the analysis fails to account for this matching. The matching variables are not included as covariates in the final model presented in Table 4. The analysis must accounts for the matching : either include the matching variables (sex, age) in the model or use a conditional logistic regression.

- Authors should consider clustering effects by village. If not accounted for, acknowledge as limitation

Results

The presentation of results in the text often merely restates percentages directly from the tables without providing a clear narrative or highlighting the most salient findings.

Discussion

- A flaw in the discussion is the repeated interpretation of non-significant findings as if they were conclusive. For example, the authors state that "persons who skipped meals...were two times more likely to develop leprosy" (Lines 343-345) and that "a low frequency of changing bed linen had higher odds of new leprosy diagnosis" (Lines 350-351). According to the authors' own analysis in Table 4, both of these variables were found to be not significant in the multivariable model. The discussion should be limited to the significant factors identified in the final adjusted model.

- A weakness of the study is the handling of temporality. By selecting participants based on their disease status and then retrospectively assessing exposures, the study cannot definitively establish that the identified "risk factors" preceded the onset of leprosy.

- Because the temporal relationship cannot be confirmed, the authors should be far more cautious in their language. They should frame their findings as associations rather than predictors or causal factors.

- The study notes that a higher proportion of cases were unemployed and experienced food shortages. The authors interpret these as risk factors related to poverty that may increase susceptibility to leprosy. However, it is equally, if not more, plausible that the debilitating physical disabilities and social stigma resulting from leprosy led to job loss and subsequent poverty, thereby causing food insecurity. In this scenario, unemployment and hunger are likely consequences, not causes, of the disease.

- A large family size is a plausible risk factor for increased transmission, however, other household factors could also be subject to reverse causality. The economic impact of leprosy on a household could lead to a deterioration in living conditions, such as the inability to maintain a home

- The link between the study's stated aims and its limitations needs to be better reconciled. The authors state they aimed to evaluate "socio-cultural" factors but then list not investigating "an elaborate social-cultural attitude" as a limitation. This suggests a disconnect between the intended scope and the actual execution of the research, which should be more directly acknowledged.

Conclusion

The conclusions and author summary should align precisely with the final, statistically significant results. The mention of "food insecurity" in the author's summary (Line 59) is not supported by the final multivariable analysis and should be removed to avoid misrepresenting the study's findings.

Reviewer #3: This version looks good to me

Reviewer #4: Strengths:Timely and context-specific, addressing leprosy in a high-burden Kenyan county. Clear case-control methodology with robust sample size.Strong ethical oversight and community engagement. Identification of actionable predictors (household size, BCG, contact patterns). Results have immediate programmatic relevance for leprosy control.

Weaknesses: Hypothesis not explicitly stated in testable terms. Stepwise regression risks overfitting and weakens interpretability. Control selection may bias results (neighbors share exposures).Over-reliance on BCG scar as proxy.

Presentation of tables could be clearer; absence of figures weakens impact.

Some causal language is overstated for observational data

PLOS authors have the option to publish the peer review history of their article (what does this mean? ). If published, this will include your full peer review and any attached files.

**Do you want your identity to be public for this peer review?** For information about this choice, including consent withdrawal, please see our Privacy Policy .

Reviewer #1: **Yes: ** Gilbert Batista

Reviewer #2: No

Reviewer #3: No

Reviewer #4: No

**Figure resubmission:**
---

## [Decision Letter · Decision Letter 2]

21 Oct 2025

Dear Karani,

We are pleased to inform you that your manuscript 'Factors Associated with New Leprosy Diagnosis in Kwale County, Kenya' has been provisionally accepted for publication in PLOS Neglected Tropical Diseases.

Best regards,

Georgios Pappas

Section Editor

Mathieu Picardeau

Section Editor

Shaden Kamhawi

co-Editor-in-Chief

Paul Brindley

co-Editor-in-Chief

Accept

Reviewer's Responses to Questions

**Key Review Criteria Required for Acceptance?**

**Methods**

-Are the objectives of the study clearly articulated with a clear testable hypothesis stated?

-Is the study design appropriate to address the stated objectives?

-Is the population clearly described and appropriate for the hypothesis being tested?

-Is the sample size sufficient to ensure adequate power to address the hypothesis being tested?

-Were correct statistical analysis used to support conclusions?

-Are there concerns about ethical or regulatory requirements being met?

Reviewer #4: Yes

**Results**

-Does the analysis presented match the analysis plan?

-Are the results clearly and completely presented?

-Are the figures (Tables, Images) of sufficient quality for clarity?

Reviewer #4: Yes

**Conclusions**

-Are the conclusions supported by the data presented?

-Are the limitations of analysis clearly described?

-Do the authors discuss how these data can be helpful to advance our understanding of the topic under study?

-Is public health relevance addressed?

Reviewer #4: Yes

**Editorial and Data Presentation Modifications?**

Reviewer #4: Accept

**Summary and General Comments**

Reviewer #4: None

PLOS authors have the option to publish the peer review history of their article (what does this mean? ). If published, this will include your full peer review and any attached files.

**Do you want your identity to be public for this peer review?** For information about this choice, including consent withdrawal, please see our Privacy Policy .

Reviewer #4: No

---

## [Editor Report · Acceptance letter]

Dear Karani,

We are delighted to inform you that your manuscript, "Factors Associated with New Leprosy Diagnosis in Kwale County, Kenya," has been formally accepted for publication in PLOS Neglected Tropical Diseases.

Best regards,

Shaden Kamhawi

co-Editor-in-Chief

Paul Brindley

co-Editor-in-Chief
